# Recommendations on Respiratory Syncytial Virus (RSV) Immunization Strategies for Infants and Young Children in Countries with Year-Round RSV Activity

**DOI:** 10.3390/vaccines14010059

**Published:** 2026-01-04

**Authors:** Fook Choe Cheah, Erwin Jiayuan Khoo, Adli Ali, Zulkifli Ismail, Rus Anida Awang, David Chun-Ern Ng, Patrick Wai Kiong Chan, Azanna Ahmad Kamar, Xin Yun Chua, Jamal I-Ching Sam, Mohd Rizal Abdul Manaf, Asiah Kassim

**Affiliations:** 1Department of Paediatrics, Faculty of Medicine, Universiti Kebangsaan Malaysia, Cheras, Kuala Lumpur 56000, Malaysia; 2Department of Paediatrics, School of Medicine, IMU University, Kuala Lumpur 57000, Malaysia; jiayuan_khoo@imu.edu.my; 3Center for Bioethics, Harvard Medical School, Harvard University, Boston, MA 02115, USA; 4KPJ Selangor Specialist Hospital, Shah Alam 40300, Selangor, Malaysia; 5KPJ Healthcare University, Nilai 71800, Seremban, Malaysia; 6Island Hospital Penang, Georgetown 10450, Penang, Malaysia; 7Department of Paediatrics, Hospital Tuanku Ja’afar, Seremban 70300, Negeri Sembilan, Malaysia; davidngce@gmail.com; 8Gleneagles Hospital Kuala Lumpur, Kuala Lumpur 50450, Malaysia; 9Department of Paediatrics, Faculty of Medicine, Universiti Malaya, Kuala Lumpur 50603, Malaysia; 10Department of Pharmacy, Hospital Canselor Tuanku Muhriz UKM, Cheras, Kuala Lumpur 56000, Malaysia; 11Department of Medical Microbiology, Faculty of Medicine, Universiti Malaya, Kuala Lumpur 50603, Malaysia; jicsam@ummc.edu.my; 12Department of Public Health Medicine, Faculty of Medicine, Universiti Kebangsaan Malaysia, Cheras, Kuala Lumpur 56000, Malaysia; 13Department of Paediatrics, Hospital Tunku Azizah, Kuala Lumpur 50300, Malaysia; asiahkassim@moh.gov.my

**Keywords:** RSV, nirsevimab, RSVpreF, maternal vaccination, long-acting RSV monoclonal antibody, seasonality, immunization

## Abstract

**Background/Objectives**: Respiratory syncytial virus (RSV) is the most common cause of lower respiratory tract infection in young children, especially during infancy, resulting in substantial morbidity and mortality. **Methods**: Acknowledging the real-world evidence on RSV immunization, the College of Pediatrics, Academy of Medicine of Malaysia, has appointed an expert panel to develop a position paper on recommendations for infant and/or maternal vaccination against childhood RSV, specifically in the Malaysian context with year-round RSV activity. **Results**: Recognizing the potential constraints and limitations in the implementation process, the expert panel recommends targeted immunization with long-acting RSV monoclonal antibody (mAb) for high-risk infants as a pragmatic first step, with subsequent scale-up to universal immunization of infants when resources permit. **Conclusions**: Immunization is the most effective strategy to prevent RSV-related lower respiratory tract infection in childhood. Year-round maternal vaccination between 28 and 36 weeks’ gestation, combined with immunization at six months for all infants, may potentially circumvent the unclear seasonality.

## 1. Introduction

Respiratory syncytial virus (RSV) infection is a major contributor to morbidity and mortality among young children. Globally, in 2019, 33 million cases of RSV-associated lower respiratory tract infection (LRTI) episodes were reported in children aged 0 to 60 months, with an estimated 3.6 million RSV-associated LRTI hospital admissions [1]. RSV infection accounted for 2% of all deaths in children aged 0 to 60 months; specifically, RSV infection contributes to 19% of RSV-related deaths within the first 6 months of life [1]. Notably, children in low-income and middle-income countries made up over 95% of RSV-related acute LRTIs and 97% of RSV-related deaths [1]. In the absence of effective therapy for RSV, prevention through immunization with monoclonal antibodies (mAbs) for infants and with maternal RSV prefusion F (RSVpreF) vaccination are proven interventions that could reduce case-related morbidity and mortality.

The advancement and real-world evidence on immunization for RSV prevention necessitate local experts’ evaluation to inform its potential implementation within the Malaysian healthcare system. This position paper, developed by an expert panel of 12 members appointed by the College of Pediatrics, Academy of Medicine of Malaysia, aims to provide recommendations on infant immunization with long-acting mAbs and maternal vaccination against RSV in Malaysia. The expert panel includes key opinion leaders in pediatric respiratory medicine, neonatology, pediatric infectious disease and immunology, microbiology, as well as public health and bioethics. The recommendations were informed by a literature search using PubMed and gray literature; cost-effectiveness analyses were incorporated where available.

## 2. Epidemiology of RSV in Malaysia

### 2.1. RSV as a Major Respiratory Pathogen in Children

Several Malaysian studies have demonstrated that RSV was among the most prevalent respiratory pathogens in infants and young children detected using respiratory samples (Figure 1) [2,3,4]. In a 27-year retrospective study (1982–2008) of hospitalized children below 5 years in Kuala Lumpur, 26.4% of respiratory samples tested positive by immunofluorescence or viral culture, of which 70.6% were RSV [2]. More recent studies from Peninsular Malaysia report similar RSV positivity rates of 15.9% (2015–2019, 23,000 cases tested via multiplex polymerase chain reaction [PCR]) [3], 17.1% (2017–2022, 4084 samples tested via direct fluorescent antibody [DFA]) [5], and 14.3% (2017–2024, 45,884 samples tested via DFA) [6]. Meanwhile, in East Malaysia, RSV-A and RSV-B were detected in 19% and 8% of 438 nasopharyngeal samples using real-time reverse PCR or real-time reverse-transcription PCR, respectively, at Sibu and Kapit Hospitals, Sarawak, over 12 months [4].

### 2.2. Age-Group Vulnerability

Younger children are more vulnerable to RSV infections, evidenced by a median age of 8 to 12 months reported across Malaysian studies [5,7,8,9]. Children under 2 years old are at the highest risk of RSV-related hospitalization, with approximately 85% of admissions recorded for this age group [3,9]. Most of these children were previously healthy, with more than 80% of hospitalized children having no documented comorbidities [8]. Although a recent local study reported no RSV-related mortality over 3 months [8], a case fatality rate of 1.6% among Malaysian children below age 5 was observed in another study conducted from 2008 to 2013 [9].

### 2.3. Lack of Immunity Against RSV and Disease

A Malaysian study reported a resurgence of RSV cases post-COVID-19 period, which increased sharply with a positivity rate of 36.3% in July–August 2022, following a sharp decline during the pandemic (8.3% in July–August 2020), surpassing pre-pandemic levels (20.6% during 2017–2019) [5]. This phenomenon is primarily attributed to immunity debt incurred during the COVID-19 period, described as decreased population immunity following an extended period of reduced exposure to circulating pathogens [10]. This raises concern as delayed RSV exposure may predispose children to more severe illness later in childhood [11], likely driven by immunological factors, including (i) lack of early-life mucosal priming [12], (ii) diminished secretory IgA and innate immune pattern recognition [13], (iii) waning of maternal antibodies in mothers who were not recently exposed to RSV [14], and (iv) the immaturity of the infant adaptive immune system at the time of first infection [15].

### 2.4. Seasonality of RSV Across Asia

Temperate countries such as China and Japan generally show well-defined peaks in the winter months [16], while non-temperate countries like Hong Kong [17,18], Taiwan [19,20], and Singapore [21] tend to experience less predictable outbreaks with residual RSV activity throughout the year (Figure 2). In Peninsular Malaysia, more pronounced infection peaks are observed either during the third quarter [3,22] or the end of the year [2,9,22,23,24]. As for East Malaysia, an earlier infection peak was observed in Sibu and Kapit, Sarawak from March to August [4]. RSV infection was independently associated with the rainy season in Kelantan (OR 3.31, 95% CI 1.44–3.69) [24]. A weak correlation between RSV infections and rainy days was seen in epidemiological studies conducted in Kuala Lumpur [2,22,23]. While RSV patterns appear to vary across Malaysia, robust surveillance is warranted to better understand local transmission trends and guide effective prevention strategies.

## 3. Disease Burden of RSV

### 3.1. Impact on Resource Utilization

RSV in younger children places a substantial strain on healthcare services (Table 1), increasing demand for hospital beds and intensive care capacity. An average monthly bed occupancy of 115% in the pediatric ward was recorded at the Kuala Lumpur Women’s and Children’s Hospital, with RSV admissions accounting for 2–22% of total admissions; meeting inpatient demand would require 82 additional beds, at an estimated cost of MYR 188.6 million (MYR 2.3 million per bed) [31]. In the same facility, children with RSV infections also required critical care more often than those with non-RSV infections (23.1% vs. 15.4%, respectively), with a significantly longer median length of stay (4 days [1–36] vs. 3 days [1–19]) [32]. Across two local studies, the need for pediatric intensive care unit (PICU) admission ranged between 15 and 15.3%, non-invasive ventilation between 10.2 and 11.2%, and mechanical ventilation between 2.5 and 6.9% [8,9]. High RSV case volumes in younger children can overwhelm hospital services and limit PICU capacity, resulting in suboptimal care.

### 3.2. Healthcare and Societal Costs of RSV Admissions

Severe RSV illness and hospitalization result in substantial expenditures (Table 1). The median direct cost of admissions for children below 5 years with acute respiratory infection at a teaching hospital in Kuala Lumpur was estimated to be USD 756 for a median hospital stay of 4 days [34]. Despite government subsidies, the median direct out-of-pocket cost remained USD 189, translating to 16.4% of monthly household income [34]. Lost parental productivity also adds to indirect costs, with each hospital admission associated with a median of three lost workdays, bringing the median societal cost to USD 871 [34]. Likely an underestimate of current actual costs, a local audit conducted three decades ago, involving children under 2 years old hospitalized for RSV infection between 1995 and 1997, reported a median admission cost of USD 358 for general inpatient care and USD 4114 for PICU care, with a median length of stay of 4 days [35,36]. These demands are significant financial burdens that could potentially overwhelm the government budget allocated for healthcare, and as such, preventive strategies should be considered to ease healthcare resource utilization.

### 3.3. High-Risk Populations for Severe RSV Disease

A disproportionate burden of RSV infection was seen among at-risk patient populations. Preterm infants under 37 weeks’ gestational age (GA) accounted for 25% of RSV-LRTI hospitalizations; among infants below 6 months, admission rates were almost fourfold for infants below 32 weeks’ GA (RR 3.87) and twofold for 32 to 37 weeks’ GA (RR 1.93) compared to all children below 2 years [37]. Local hospital admission data on RSV reported a substantial proportion who were extremely preterm infants (26 to 28 weeks’ GA), with nearly half (45%) requiring PICU admission at the Kuala Lumpur Women’s and Children’s Hospital [38]. Meanwhile, the FLIP-2 Spanish prospective study involving hospitalized premature infants (32 to 35 weeks’ GA) reported that 17.8% were admitted to the ICU and 7.4% required mechanical ventilation [39].

Infant risk stratification for RSV hospitalization is a subject of interest, especially from an economic perspective in resource-limited settings, as it helps prioritize prophylaxis for those at the highest risk of severe disease. The risk factors for severe RSV infection leading to hospitalization are outlined in Box 1.

Box 1Risk factors for severe RSV infection leading to hospitalization.
Gestational ages ≤ 35 weeks [39,40];Congenital heart disease [37,40];Chronic lung disease and bronchopulmonary dysplasia [37,40,41];Congenital anomalies or syndromic infants (e.g., Down’s Syndrome) [37];Neuromuscular disorders [40,42];Immunodeficiency disorders [40,41].


Prior to the recommendation of universal immunization against RSV for all infants, palivizumab prophylaxis for preterm infants below 35 weeks’ gestation was the primary preventive strategy in many countries [43]. In Malaysia, while the Pediatric Pharmacy Services Guideline advises administering palivizumab to infants with chronic lung disease or a history of prematurity (< 35 weeks’ GA) [44], the *Universiti Kebangsaan Malaysia* teaching hospital reserves palivizumab for preterm neonates < 29 weeks’ GA, weighing 1000 g and below, and/or diagnosed with bronchopulmonary dysplasia [45]. Although some settings have secured budget allocation for infants at the highest risk of RSV disease [45], most Malaysian public hospitals administer palivizumab on an ad hoc basis without specific funding, reflecting the absence of a coordinated national policy.

### 3.4. Long-Term Clinical Sequelae Following RSV Infection

In children, clinical manifestations of RSV infection can range from mild respiratory symptoms to severe illness with acute and long-term consequences. That said, non-medically attended mild RSV infections can also have persistent symptoms beyond 15 days in half (50.5%) of healthy, term infants in the form of rhinitis (99%), cough (96.9%), and wheezing (66%) [46]. This can impose a considerable social burden among parents, seen as impairment in usual daily activities (in 59.8% of episodes), worries (75.3%), anxiety (34%), and work absenteeism (10.8%) [46]. In more serious cases, infants who experience RSV bronchiolitis within their first 6 months of life have around 30% higher odds of developing pneumonia and otitis media, as well as requiring antibiotics in the following 6 months [47]. Early-life RSV LRTI can have long-term respiratory sequelae, including recurrent wheezing [48], asthma [49], abnormal lung function [50], and post-infection bronchiolitis obliterans [51]. Overall, RSV infections can result in substantial morbidity, often requiring medical attention in the long term (Table 2).

## 4. The Virus, Mechanism of Disease, and Immune Defenses

RSV (Figure 3) is a single-stranded RNA *orthopneumovirus* of the *Pneumoviridae* family, comprising two major subtypes—RSV-A and RSV-B [52]. Following inhalation, RSV infects the upper airway epithelium and later spreads to the bronchioles of the lower respiratory tract, where viral replication is enhanced [53]. For viral entry into the host cells, the attachment glycoprotein (G) binds to the cell surface, while the fusion glycoprotein (F) facilitates membrane fusion [52]. The RSV F protein then inhibits interferon-λ production, leading to continuous viral infection that becomes amplified [54]. This is accompanied by airway damage largely driven by an immune-mediated response [54]. If confined to the upper airways, RSV infection typically presents with symptoms of an upper respiratory infection; however, in previously unexposed infants, the virus often spreads to the lower respiratory tract, causing LRTI [53].

Younger children are more vulnerable to severe illnesses from airway obstruction due to their smaller airways, reduced respiratory capacity, and lower respiratory reserve [53]. Additionally, protection against RSV infection in infants relies on maternal antibody levels, which wane rapidly after birth and are mostly absent at 6 months old [59]. These children are also susceptible to RSV reinfections, demonstrating a primary infection rate of 86% and a reinfection rate of around 35% in their first 3 years of life [59]. RSV reinfections could be attributed to short-lived primary RSV infection-induced antibody response, compounded by the immune system’s limited ability to develop efficient protective immunity against the virus, a rapid decline in antibody levels, and antigenic variations of RSV strains in subsequent epidemic seasons [59].

### Protection Against RSV Disease with Vaccination

Early efforts to develop an RSV vaccine for infants were historically associated with safety concerns about formalin-inactivated vaccines causing enhanced respiratory disease [60]. Nonetheless, advances in the understanding of RSV structural biology have marked a turning point in vaccine development, specifically the discovery of the RSV F glycoprotein as a highly conserved antigen that refolds from a metastable prefusion form to a stable postfusion form during viral entry, making it an ideal candidate for passive prophylaxis [56]. This progress led to pivotal milestones in RSV prevention in the 2020s (Figure 4), with the approval of subunit vaccines for older adults and maternal RSV vaccination to protect infants via transplacental antibody transfer [61,62]. Currently, other RSV vaccine candidates in clinical development raise hopes for longer-term protection against this disease [52].

Immunization against RSV is recommended to protect individuals at the “extremes of ages”, who are at the highest risk of severe disease. Infants are protected either through maternal vaccination during pregnancy or infant immunization with mAbs, and older adults through RSV vaccination [72]. These immunization strategies protect infants during early life, when the risk of morbidity and mortality from RSV infection is greatest among children below 2 years of age [3,73,74]. Both maternal vaccination and mAbs have shown robust antibody responses against RSV-A and RSV-B [74,75], supporting broad coverage against circulating strains. As the subunit-type maternal vaccines and mAbs do not confer long-term protection, populations at risk will need to be re-immunized if necessary. Achieving herd immunity against RSV is unlikely with this strategy because of waning immunity and viral antigenic drift [76]. Nevertheless, population-level immunization remains crucial in reducing severe disease among vulnerable age groups [76].

## 5. Current RSV Immunization Approaches

### 5.1. Infant Passive Immunization

In June 1998, palivizumab became the first mAb approved by the US FDA for the prevention of RSV in high-risk children [65]. Infants who meet the criteria for prophylaxis may receive up to five monthly doses of palivizumab during their first RSV season, and at the start of their second season if indicated [77]. This schedule is based on its half-life of 20 days [78], which confers an estimated protection period of 28 days per dose [79]. A 2021 Cochrane review determined that palivizumab prophylaxis significantly reduces RSV-related hospitalization in high-risk children (RR 0.44, 95% CI 0.3–0.64) [80]. Nonetheless, its short duration of action, the need for repeated monthly doses, and its high cost support the recommendation for targeted immunization of high-risk infants to reduce severe disease in a cost-effective manner [81].

Nirsevimab, the first long-acting mAb, was introduced in July 2023 to prevent RSV-associated LRTI in infants during their first RSV season and in high-risk children up to 24 months during their second RSV season [67,82]. With a half-life of 71 days, nirsevimab provides almost immediate protection after a single dose, which lasts for at least 5 months [78]. Its efficacy has been established in multiple pivotal trials (Table 3) [83,84,85,86,87], with a meta-analysis of 45,238 infants reporting a pooled efficacy of 88.4% (95% CI 84.7–91.2) against RSV-related hospitalization [88]. The MEDLEY trial found that nirsevimab has a safety profile comparable to palivizumab [89]; there is also some evidence that RSV protection could linger for longer, with post hoc analysis reporting nearly tenfold higher and more sustained RSV-neutralizing antibody levels up to 1 year [78].

The availability of nirsevimab has marked a paradigm shift in RSV prevention from targeted immunization of high-risk infants to universal immunization of all infants [90]. The NIRSE-GAL study in Galicia, Spain, achieved 92% coverage, demonstrating 70.7% (95% CI 42.4–85.1) effectiveness against RSV-related LRTI hospitalization and 80.3% (54.6–91.5) effectiveness in preventing RSV-related LRTI hospitalization requiring oxygen support during the 2023–2024 season, with no new safety signals identified [91]. Similarly, Chile’s NIRSE-CL study achieved a coverage of 94% nationwide, resulting in an effectiveness of 76.4% (72.6–79.7) against RSV-related LRTI hospitalization and 84.9% (79.5–88.9) against ICU admissions [92]. Meanwhile, the REVIVE study conducted in Western Australia reported an adjusted effectiveness of 88.2% (73.5–94.7) against RSV-associated acute respiratory infection hospitalizations over 7 months [93]. The consistent effectiveness observed across real-world studies highlights the public health value of universal infant immunization with nirsevimab.

A new mAb, clesrovimab, was recently approved in June 2025 for preventing RSV LRTI in infants during their first RSV season [68]. Despite its shorter half-life compared to nirsevimab (44 days vs. 71 days), durable protection was observed across the typical five-month RSV season [94,95]. In the phase 2b/3 CLEVER trial (MK-1654-004), clesrovimab reduced RSV LRTIs at 150 days post-dose in healthy preterm and full-term infants from birth to 1 year (efficacy 60.4%, 95% CI 44.1–71.9); even greater efficacy was observed in preventing LRTI hospitalizations (84.2%, 66.6–92.6) [94,96]. Additionally, the phase 3 SMART trial (MK-1654-007) indicated that clesrovimab is well tolerated in high-risk infants, with a safety profile and RSV incidence rates comparable to monthly palivizumab up to 150 days [95].

### 5.2. Maternal Vaccination

The RSVpreF vaccine is indicated for maternal active immunization aimed at preventing RSV-associated LRTI in infants below 6 months old [97,98,99]. Although the European Medicines Agency approval permits use from 24 to 36 weeks’ gestation as per the MATISSE trial [99,100], slightly higher rates of preterm birth among RSVpreF recipients, though not statistically significant, perhaps led the US FDA to limit its use to 32 to 36 weeks’ gestation in view of this potential risk [101,102]. Geographical variation was evident, with the vaccinated groups reporting a higher relative risk of preterm birth in Argentina and South Africa (significant in South Africa) [103]. A single RSVpreF dose between 32 and 36 weeks’ gestation was also implemented in Canada and Argentina [104,105], while the UK and Australia recommend administration from 28 to 36 weeks [106,107].

Real-world data from Argentina’s BERNI study showed that maternal RSVpreF vaccination was effective against RSV-related LRTI hospitalization from birth to 3 months (78.6%, 95% CI 62.1–87.9), with protection sustained up to 6 months (71.3%, 53.3–82.3) [105]. A modelling study in Australia suggests that achieving 70% year-round coverage of maternal RSVpreF vaccination could reduce infant hospitalization under 3 months by 60% [108]. In Malaysia, the 2024 Malaysian Maternal Immunization Consensus Guidelines position RSVpreF as a highly efficacious vaccine and recommend its administration between 32 and 36 weeks’ gestation [109].

The maternal vaccination approach appears to be promising in the local context. In Malaysia, an example of maternal vaccination uptake can be observed with the tetanus, diphtheria, and pertussis (Tdap) vaccine, where the government formally introduced Tdap for pregnant women in the National Immunization Program and announced free provisions in government facilities from 2024 onwards [110]. While official data on its uptake rate has yet to be published, a small study of 80 Malaysian pregnant or new mothers conducted in early 2024 reported that 33% of new mothers have been administered the Tdap vaccine, while 63% of pregnant mothers have either received or intend to receive it [111]. With the inclusion of the Tdap vaccine in the Malaysian Maternal Immunization Consensus Guidelines by the Obstetric and Gynecological Society of Malaysia, released in late 2024 [109], Tdap uptake is expected to increase further. A similar outcome may be expected if maternal RSV vaccination were to be included in the program in the future.

### 5.3. Infant Passive Immunization vs. Maternal Vaccination

Both infant mAb immunization and maternal RSVpreF vaccination offer distinct benefits and limitations, though no direct comparative data currently exist on their effectiveness in preventing RSV-associated LRTI in infants. Dosage and administration of infant mAb and maternal RSVpreF are outlined in Appendix A. Immunization with mAb provides protective antibodies directly to the infant, unaffected by the variability in maternal response and transplacental transfer [101].

While maternal vaccination may offer immediate protection at birth and could be less susceptible to F protein mutations, its effectiveness might be compromised if antibody production or placental transfer is suboptimal, especially in immunocompromised mothers or if the infant is born prematurely within 14 days of vaccination [101]. RSVpreF induces robust immune responses in pregnant individuals, resulting in high RSV-neutralizing titers in their newborns [112]. However, there is currently insufficient evidence on how sustained the antibodies in subsequent pregnancies are and whether they protect the pregnant individuals themselves against RSV infections.

## 6. Cost-Effectiveness of RSV Immunization Approaches

Evaluating the cost-effectiveness of RSV prevention strategies in both targeted and universal approaches is essential to inform policymaking and optimize implementation. Table 4 outlines published cost-effectiveness analyses of RSV prevention strategies.

### Cost-Effectiveness for RSV Prevention in Malaysia

To support the expert panel’s deliberation, the study group performed a focused cost-effectiveness analysis of RSV prevention strategies in the Malaysian healthcare context. A decision tree model was used to account for direct and indirect medical costs related to RSV hospitalization and mortality.

The findings from the cost-effectiveness analysis indicate that nirsevimab immunization in high-risk infants prior to hospital discharge is the most affordable and costs substantially less than palivizumab, at almost one-tenth of its price. Nirsevimab alone or in combination with maternal vaccination is cost-effective compared to palivizumab; dominant strategies that are more cost-effective than palivizumab include (i) nirsevimab for high-risk infants, (ii) maternal vaccination with complementary nirsevimab for infants unprotected by maternal vaccination, and (iii) maternal vaccination with complementary nirsevimab plus extended nirsevimab dosing for high-risk infants at 6 and 12 months. Nirsevimab for high-risk infants achieved the highest spending-to-savings ratio of 1:1.62, followed by maternal vaccination with complementary nirsevimab plus extended nirsevimab dosing for high-risk infants of 1:1.28. Further details of the cost-effectiveness analysis are outlined in Appendix B.

## 7. Ethics, Equity, and Feasibility of RSV Immunization

Historically, palivizumab has been reserved for high-risk infants due to cost issues, thereby excluding those at moderate risk and perpetuating preventable inequities in protection. The advent of new long-acting mAbs and maternal RSV vaccines now marks a critical ethical inflection point [119]. The true ethical imperative is not merely offering cheaper protection, but the systemic commitment to a universal access model, thereby dismantling the inequitable risk-stratification paradigm entirely. Policy discussions must pivot from cost-containment relative to the “palivizumab era” to assessing the absolute program affordability required for universal deployment, acknowledging that even reduced per-course costs translate into significant, politically challenging budgetary demands on national health systems.

The concurrent emergence of maternal vaccination and infant immunization strategies also raises uncertainties about which is the ethically preferable option beyond cost—should protection be delivered directly to infants via long-acting mAbs or indirectly via maternal vaccination? Long-acting mAbs may add to an already crowded pediatric immunization schedule, potentially affecting adherence and raising concerns about pain, distress, and trust in healthcare services. Conversely, maternal vaccination avoids invasive procedures for the child but shifts the decision to the pregnant individual, invoking considerations of her own pain and distress, alongside parental responsibility and individual autonomy.

While existing national cold chain infrastructure is cited as supportive [120], potential limitations in system capacity and distribution topology can hinder the implementation of RSV prevention strategies. Maternal vaccination necessitates expanding cold chain access into antenatal clinics, while infant mAb delivery demands sufficient storage and seamless integration into community pediatric clinics. The challenge of achieving supply chain resilience extends particularly to last-mile delivery and comprehensive product tracking, which must bridge the infrastructural and administrative gaps between state-managed systems (e.g., National Immunization Program) and the often-decentralized private healthcare market, a divide common across diverse healthcare settings in countries with year-round RSV activity.

Healthcare workers should also be adequately trained, not only in clinical areas such as efficacy, contraindications, and adverse events, but also in operational competencies, including workflow integration and patient counselling. Additionally, parental acceptance remains crucial; as hesitancy is rarely resolved by simple “education”, public engagement must move beyond basic information, proactively addressing the unique skepticism surrounding a novel biologic through transparent dialogue and acknowledging the socio-political and experiential roots of distrust. Clinical efficacy must be actively validated and matched by deliberate, strategic efforts to build and maintain robust public trust.

## 8. Final Considerations and Position Statements

RSV prevention strategies should consider feasibility, administrative capacity, and budget constraints (Figure 5). With these considerations in mind, the expert panel advises targeted immunization with long-acting mAbs for high-risk infants prior to hospital discharge as the pragmatic first step in the implementation of a national childhood RSV protection program; immunization may be repeated during the second year of life for eligible at-risk infants.

Subsequent scale-up of the program to universal administration to protect all infants warrants serious consideration when resources permit. This can be achieved either through maternal vaccination with complementary long-acting mAbs or universal use of long-acting mAbs for all infants. As maternal vaccination is substantially less costly than long-acting mAbs, it may be worthwhile to explore the implementation of maternal vaccination for all mothers, especially in resource-limited settings.

## 9. Conclusions

Although real-world evidence shows the effectiveness of long-acting mAbs for RSV prophylaxis as a seasonal strategy [91,92], the absence of clear seasonality in Malaysia is a key limitation that complicates the optimization of immunization schedules for maximum protection. Therefore, the expert panel recommends addressing the issue of unclear seasonality with a combined strategy of maternal vaccination to protect infants from birth through 6 months, followed by universal infant immunization with long-acting mAbs at 6 months to provide seamless protection across the first year of life. This strategy avoids potential interference that may occur with the coadministration of other childhood vaccinations given at birth. Taken together, this addresses the high burden of disease in early infancy and the continued RSV circulation observed year-round (with peak incidence between 8 and 12 months of age) in a country with unclear RSV seasonality.

## Figures and Tables

**Figure 1 vaccines-14-00059-f001:**
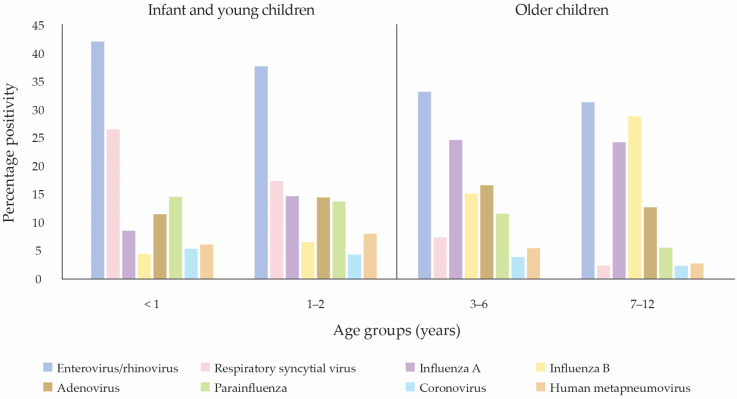
Percentage positivity of respiratory viruses causing acute respiratory infection among children in Malaysia, adapted from Low et al. (2022) [3].

**Figure 2 vaccines-14-00059-f002:**
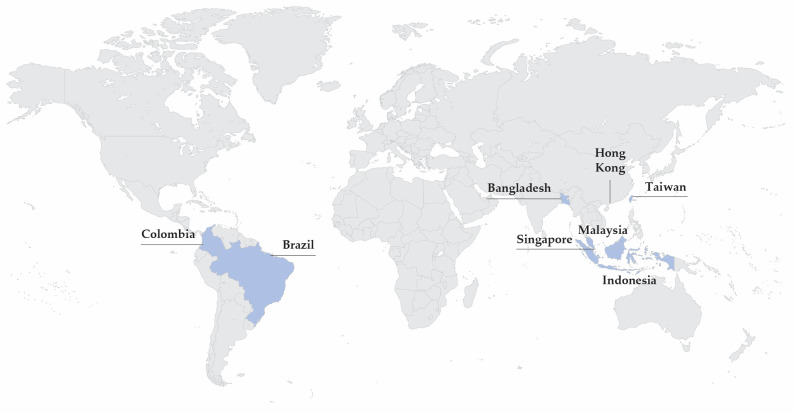
Countries with year-round or variable RSV seasonality, based on published epidemiological studies [2,3,9,16,17,18,19,20,21,22,23,24,25,26,27,28,29,30].

**Figure 3 vaccines-14-00059-f003:**
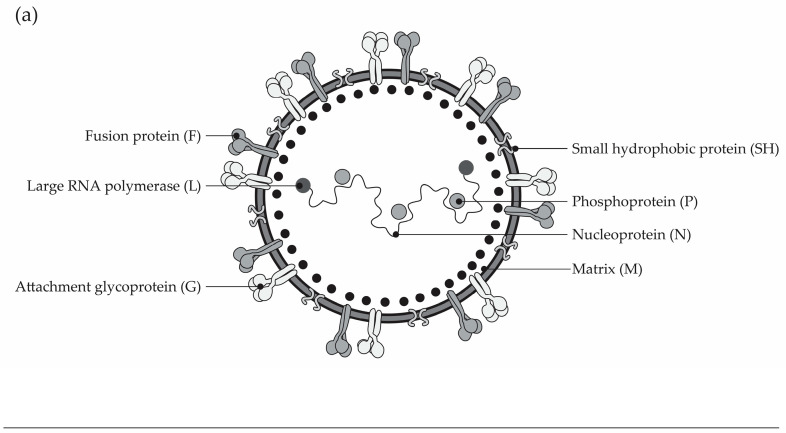
(**a**) RSV structure and the (**b**) RSV F glycoproteins, modified from Akagawa et al. (2024) [55]. The RSV F glycoprotein is integral in the entry of the virus into cells; during viral entry, the RSV F glycoprotein refolds from a prefusion to postfusion conformation, driving fusion of the viral and host membranes and enabling viral entry, after which viral replication proceeds [56]. Passive prophylaxis targets several of the six sites on the RSV F glycoprotein: palivizumab, site II [57]; nirsevimab, site Ø [57]; clesrovimab, site IV [58].

**Figure 4 vaccines-14-00059-f004:**
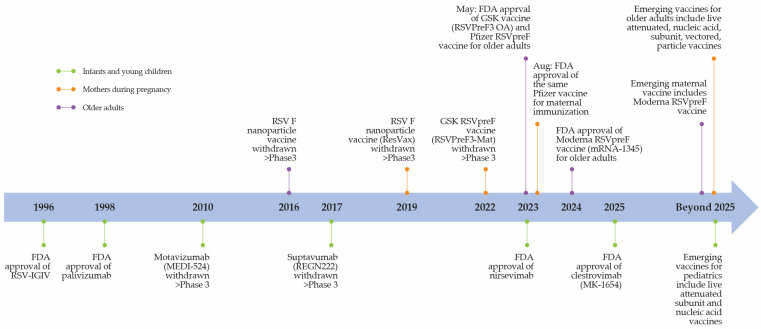
Evolution of RSV prevention and immunization strategies [61,62,63,64,65,66,67,68,69,70,71].

**Figure 5 vaccines-14-00059-f005:**
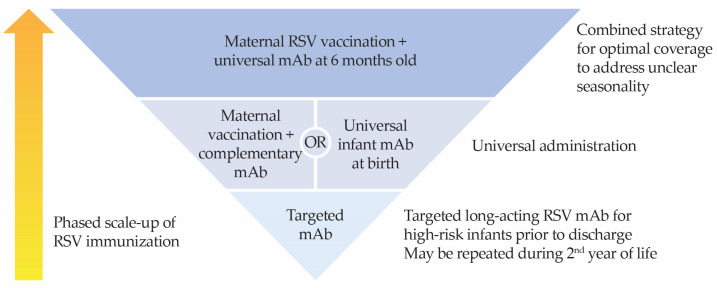
Proposed RSV prevention strategies for the Malaysian healthcare system. Complementary nirsevimab refers to nirsevimab given to infants whose mothers have not received RSV vaccination during pregnancy, have unknown vaccination status, or have delivered within 14 days of maternal vaccination.

**Table 1 vaccines-14-00059-t001:** Healthcare resource utilization and costs associated with RSV infection [8,9,31,32,33,34].

Healthcare Resource Utilization	Costs		
	Direct (Medical)	Direct (Non-Medical)	Indirect
Medically attended visits Hospital admissions Prolonged length of stay Increased hospital bed occupancy Intensive care unit care Mechanical ventilation Non-invasive ventilation	Clinical management costs Hospitalization Medications Diagnostic investigations	Transport Accommodation/lodging	Parental income loss Loss of work productivity

**Table 2 vaccines-14-00059-t002:** Clinical manifestations of RSV infection [46,47,48,49,50,51].

Acute Symptoms	Long-term Respiratory Sequelae
Cough Fever Feeding difficulties Rhinitis Shortness of breath Wheezing Vomiting	Recurrent wheezing Asthma Abnormal lung function Post-infection bronchiolitis obliterans

**Table 3 vaccines-14-00059-t003:** Efficacy outcomes from pivotal trials of nirsevimab.

Trials	RSV MA-LRTI	RSV LRTI Hospitalization	Very Severe RSV MA-LRTI	Very Severe RSV LRTI Hospitalization
Phase 2b [83] (N = 1453; nirsevimab = 969)	70.1% (52.3–81.2)	78.4% (51.9–90.3)	-	-
MELODY, primary cohort [84] (N = 1490; nirsevimab = 994)	74.5% (49.6–87.1)	62.1% (−8.6–86.6)	-	-
MELODY, all subjects [85] (N = 3012; nirsevimab = 2009)	76.4% (62.3–85.2)	76.8% (49.4–89.4)	78.6% (48.8–91)	-
HARMONIE, through RSV season (~3 months) [86] (N = 8058; nirsevimab = 4037)	-	83.2% (67.8–92)	-	75.7% (32.8–92.9)
HARMONIE, through 180 days [87] (N = 8058; nirsevimab = 4037)	-	82.7% (67.8–91.5)	-	75.3% (38.1–91.8)

Data presented as efficacy (95% CI). LRTI: lower respiratory tract infection; MA: medically attended; RSV: respiratory syncytial virus.

**Table 4 vaccines-14-00059-t004:** Cost-effectiveness analyses of RSV prevention strategies from other countries.

Source	Strategy	Locality	Perspective	Price	WTP Threshold	ICER
Zeevat et al. (2025) [113]	Universal nirsevimab (year-round)	Netherlands	Societal	EJP: EUR 220	EUR 50,000/QALY	EUR 122,478/QALY (vs. palivizumab)
	Universal nirsevimab (seasonal + catch-up)					EUR 50,000/QALY (vs. palivizumab)
Langedijk et al. (2025) [114]	Universal nirsevimab	Netherlands	Societal	RSVpreF: EUR 180 Nirsevimab: EUR 547.3 (assumed)	-	EUR 592,404/QALY (vs. no intervention)
	Maternal RSVpreF + complementary nirsevimab					EUR 329,187/QALY (vs. no intervention)
Hutton et al. (2024) [115]	Universal nirsevimab (seasonal + catch-up)	US	Societal	USD 445 *	-	Universal: USD 153,517/QALY (vs. no intervention) High-risk children in second season: USD 308,468/QALY (vs. no intervention)
Wang et al. (2025) [116]	Universal nirsevimab (seasonal)	China	Societal	USD 263.83	USD 26,866	USD 13,073.79 (vs. no intervention)
	Universal nirsevimab (year-round)					USD 24,323.26 (vs. no intervention)
Noto et al. (2025) [117]	Universal nirsevimab	Japan	Payer Societal	JPY 45,000 EJP: JPY 45,496	JPY 5,000,000	Payer: JPY 4,537,256/QALY (vs. palivizumab) Societal: JPY 1,695,635/QALY (vs. palivizumab)
Gebretekle et al. (2024) [118]	Targeted nirsevimab for at-risk infants (seasonal + catch-up)	Canada	Health system and societal	Nirsevimab: CAD 110–190 RSVpreF: CAD 60–125	-	Infants at moderate/high-risk: CAD 27,891/QALY (vs. palivizumab)
	Year-round maternal RSVpreF + nirsevimab for high-risk infants				CAD 50,000/QALY	CAD 204,621/QALY (vs. seasonal nirsevimab for infants at moderate/high risk)
	Universal nirsevimab (seasonal + catch-up)					CAD 512,265 (vs. year-round maternal RSVpreF + nirsevimab for high-risk infants)

* Assumption: With doses purchased through Vaccines for Children (USD 395) and commercial channels (USD 495), a weighted average of USD 445/dose was used as a base-case cost. EJP: economically justifiable price; ICER: incremental cost-effectiveness ratio; QALY: quality-adjusted life-years; RSVpreF: RSV prefusion F; WTP: willingness-to-pay.

## Data Availability

The authors confirm that the data supporting the findings of the cost-effectiveness analysis are available in Appendix B. Additional raw data that supports the findings of this report are available from the corresponding author upon reasonable request.

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
