# Peer review of "Recommendations on Respiratory Syncytial Virus (RSV) Immunization Strategies for Infants and Young Children in Countries with Year-Round RSV Activity"

_vaccines, 2026, doi:10.3390/vaccines14010059_

Round 1

Reviewer 1 Report

Comments and Suggestions for Authors

A manuscript devoted to the analysis of RSV infection in infants and children up to 5 years old living in Asian countries was submitted to Vaccines by Fook Choe Cheah et al.

The authors examine RSV in considerable details as an infectious agent affecting newborns and children up to 5 years old, describing known cases in the literature where RSV was the primary cause of death in such patients.

The reviewer believes that the manuscript's topic and quality meet Vaccines' requirements; however, the text lacks summary tables that would allow the reader to quickly review the essence of the topic and provide valuable references. The authors also do not emphasize the importance of vaccination in preventing the disease in children, which is surprising given the specific nature of the Vaccines journal.

The reviewer believes that the authors should highlight the role of vaccination and try to structure some of the material to tables and/or graphs and diagrams to illustrate the manuscript.

Reviewer 2 Report

Comments and Suggestions for Authors

Cheah et al describe the problem of RSV infection at a very high level of scientific understanding. In addition, the possible economic and sociological aspects of the varies preventive measures are well described.

For this reviewer it seems also clear that the vaccination of mothers is the most effective economical approach. Maybe some issues might be discussed further on this vaccination route. While RSVA and RSVB coexist throughout RSV seasons, RSVA is more prevalent, fatal, and epidemic-prone in several countries, including China but perhaps also in Malaysia. Vaccination of mothers induces poly - clonal antibodies that may be more efficient to control varies RSV strains than monoclonal antibodies given to newborns. In addition, the current strains in Malaysia might induce a boost to (vaccinated) pregnant mothers.

What is discussed by Cheah et al is the public, sociological acceptance of vaccinations of pregnant mothers. Is the skepsis against vaccination high in Malaysia? A general vaccination of the population might enhance the herd immunity. Is this an option?

It is well known that newborns or even preterm babies not protected by maternal antibodies are at high risk to get infected by RSV. At this point passive antibodies proposed are very effective. Vaccination of children over one or two years of age are not favored. The (partly) exclusion of this option is not entirely clear to this reviewer.

Round 2

Reviewer 1 Report

Comments and Suggestions for Authors

Authors made the changes reqiured by reviewers. Manuscript might be recommended for the acceptance in its current form after a minor revision

1) Figure 3 in not a Figure

Sincerely,

Author Response

Comment 1: Figure 3 in not a Figure. 

Response: Thank you for your comment, we have amended this to Table 1 (see Page 4, line 143).